# Development of Polymer Composites Using Surface-Modified Olive Pit Powder for Fused Granular Fabrication

**DOI:** 10.3390/polym16212981

**Published:** 2024-10-24

**Authors:** Pedro Burgos Pintos, Mirko Maturi, Alberto Sanz de León, Sergio I. Molina

**Affiliations:** Departamento Ciencia de los Materiales, Ingeniería Metalúrgica y Química Inorgánica, IMEYMAT, Facultad de Ciencias, Universidad de Cádiz, Campus Río San Pedro, s/n, 11510 Puerto Real, Spain; pedro.burgos@uca.es (P.B.P.); mirko.maturi@uca.es (M.M.); sergio.molina@uca.es (S.I.M.)

**Keywords:** large format additive manufacturing, fused granular fabrication, olive pit, agro-waste valorization, bio-based composites, surface modification

## Abstract

In this study, olive pit agro-waste from the olive oil industry is valorized by incorporating it as an additive in a polyethylene terephthalate glycol (PETG) matrix to develop bio-based composite materials for large format additive manufacturing (LFAM). The olive pits were first ground into olive pit powder (OPP) and then functionalized by polymerizing poly(butylene adipate-co-terephthalate) PBAT on their surface, resulting in a hydrophobic, modified olive pit powder (MOPP) with enhanced compatibility with the PETG matrix. OPP and MOPP composites were compounded and 3D-printed via Fused Granular Fabrication (FGF) using 5, 10, and 15 wt.% concentrations. The PBAT coating increased the degradation temperature and specific heat capacity of the material, contributing to a lower melt viscosity during printing, as confirmed by MFR, MDSC, and TGA analyses. Tensile testing revealed that MOPP composites generally exhibited superior mechanical properties compared to OPP composites, likely due to the improved compatibility between PBAT on the MOPP surface and the PETG matrix. SEM analysis further validated these findings, showing a highly irregular and porous fracture surface in OPP composites, while MOPP composites displayed a smooth surface with well-integrated MOPP in the PETG matrix.

## 1. Introduction

Additive Manufacturing (AM) technologies allow for the manufacture of customizable products with high formal complexity, thanks to its layer-by-layer deposition system, differing from conventional manufacturing processes due to their efficient use of materials and minimal waste generation [1]. Among the different AM technologies, Large Format Additive Manufacturing (LFAM) stands out in different industrial sectors like automotive, naval, aerospace, and construction [2]. Within LFAM, extrusion-based methods, Fused Granular Fabrication (FGF), is probably the most popular because it allows for higher deposition rates of material and the processing of a wide range of polymers and polymer-based composites. FGF uses pellets as feedstock, which also reduces raw material costs by a factor of 10 when compared to filament-based AM technologies [3]. This has numerous advantages compared to classical manufacturing technologies, such as lower tooling costs and enhanced manufacturing flexibility [4]. Engineering polymers [5] and fiber-reinforced polymer composites [6] are widely used in LFAM for their excellent mechanical properties during service. However, they are all derived from fossil fuels, and their production involves a high carbon footprint [7].

The development of composite materials obtained by the incorporation of agro-waste into engineering polymer matrices aims to reduce the amount of fossil fuel-derived content and improve the sustainability of products [8]. This agro-waste can be obtained in the form of fibers or particles, depending on the production process that generated them [9,10,11]. On the Mediterranean coast, particularly in Spain, some of the most produced agro-waste is cork and olive residues due to the natural presence of oaks and olive trees in this region. The olive industry and the manufacture of olive pits (OP) generate millions of tons of these residues annually [12,13], which can be used as reinforcement in composites due to their high hardness. For this, the OP is typically pulverized first to obtain a micron-sized olive pit powder (OPP) [10,11]. However, OPP exhibits low compatibility with polymer matrices. OPP is mainly composed of lignin, cellulose, and hemicellulose, and their hydrophilicity causes this lack of compatibility with engineering polymers, which are generally hydrophobic [14]. Therefore, composites produced by mixing engineering polymers with pristine OPP have generally poor mechanical properties, as the polymer matrix is not able to transfer mechanical stress to the filler which cannot contribute to the overall mechanical resistance of the composite. The filler does not integrate well within the matrix, leading to material porosity or acting as a crack initiator, which limits the amount of additive that can be incorporated into the polymer matrix [15].

It is, therefore, of great interest for the materials science community to develop surface-modification strategies that can improve the compatibility of agro-waste with polymer matrices to maximize the mechanical properties of the final composite. Traditionally, this is performed by the reactive extrusion of the polymer matrix and the agro-waste with a compatibilizer. The latter is usually composed of a thermoplastic polymer functionalized by maleic anhydride (MA), which can react with the surface hydroxylated groups of the biomaterial, binding it to the polymer matrix and improving the matrix-to-filler load transfer. However, this approach causes the crosslinking of the polymer structure, leading to increases in the stiffness of the composite at the expense of its ductility [12,16,17].

In order to address these limitations, surface-initiated polymerization methods have recently gained attention in the field of nanocomposite materials. This approach involves growing or attaching polymeric chains to the surface functional group of the filler. The choice of the polymer to be grown on the filler’s surface is determined based on evaluating the potential intermolecular interactions between the polymer and the polymer matrix. For example, it has been demonstrated that the grafting of polymer chains from or to the surface of the nanofiller improves the filler–matrix interface of carbon nanofibers [18], graphene nanoplatelets [19], nanocellulose [20], or other bio-based materials [21].

Taking all this into account, in this work, we have developed a series of thermoplastic polymer composites using PETG as the polymer matrix and OPP as the filler, with loadings ranging from 5 to 15 wt.% for LFAM applications. To increase the compatibility of OPP with the matrix, OPP was chemically modified by polymerization of surface-grafted poly(butylene adipate-*co*-terephthalate) (PBAT), obtaining a modified olive pit powder (MOPP). It was observed that this modification improved the thermal stability of the material by altering its specific heat capacity (C_p_) and reducing its degradation during the processing, which led to better printability. This resulted in improved mechanical properties of the MOPP composites compared to those with OPP. The enhanced compatibility of OPP with the matrix was assessed by SEM, which showed a better integration of the MOPP within the PETG matrix and a significant reduction in the porosity of the material.

## 2. Materials and Methods

### 2.1. Materials

Polyethylene terephthalate glycol (PETG) Mimesis DP 300 was supplied in pellets by Selenis (Portalegre, Portugal). The olive pit powder (OPP) was kindly supplied by Prof. La Rubia from the University of Jaén (Jaén, Spain). This olive pit was obtained by grinding previously dried olive pits in a cryogenic mill, obtaining particle sizes between 32 µm and 250 µm [12]. All chemicals were purchased from Sigma Aldrich (St. Louis, MO, USA) and used as received.

### 2.2. Synthesis of PBAT-Modified Olive Pit Powder (MOPP)

The synthesis of MOPP was carried out in three steps. First, 150 g of OPP was dispersed under vigorous magnetic stirring in a solution composed of 150 g of chloroacetic acid in 1 L of water in a 3 L beaker. Then, the reaction was allowed to take place by adding a second solution composed of 150 g of NaOH in 1 L of water. The mixture was vigorously stirred for 15 min at room temperature, followed by sonication for a further 15 min. At this point, the reaction was quenched by pouring the mixture into 1 L of isopropanol. The product was collected by filtration and consecutive washings, first with water, then isopropanol. In the second step, carboxylated OPP from the previous step was placed in a 2 L round-bottomed flask equipped with a magnetic stirrer and dropping funnel under an Ar atmosphere, and it was dispersed in 1.5 L of methanol. Then, 75 mL of 98% sulfuric acid was slowly added dropwise under vigorous stirring, and the mixture was refluxed for 1 h to ensure efficient esterification. Then, the mixture was cooled to room temperature, poured into a beaker, and neutralized by slowly adding a saturated NaHCO_3_ solution until bubbling stopped occurring. The esterified product was isolated by filtration and then washed twice with water and twice with acetone. Then, in a flame-dried 1 L round-bottomed flask, OPP methyl ester (115 g), dimethyl adipate (45 mL, 0.274 mol), dimethyl terephthalate (53.2 g, 0.274 mol), 1,4-butanediol (48.3 mL, 0.548 mol), and dibutyltin (IV) oxide (2 g, 8 mmol) were sequentially added. The mixture was heated at 180 °C for 1 h, during which the produced methanol was separated by distillation. Then, the vacuum was applied for 1 h to drive the polymerization reaction. After cooling to RT, the solid mixture was poured into 1 L of methanol and filtered. The obtained solid was cleaned three times by repeated redissolution in the minimum amount of DCM and re-precipitation in methanol. The overall synthesis produced 164 g of MOPP.

### 2.3. Processing

A scheme illustrating the manufacturing of the OPP and MOPP composites is presented in Figure 1. PETG, OPP, and MOPP were dried for at least 6 h at 60 °C to remove any residual moisture. Then, neat PETG with 5, 10, and 15 wt.% OPP and MOPP were processed in a Scamex Rheoscam D20 (Isques, France) working at 72 rpm with a L:D of 47 and a temperature profile from the feeding hopper to the nozzle of 205-210-210-210-210 °C was used for the synthesis of all the composites. In all cases, a continuous filament was obtained and cut into small pieces of 4–5 mm length using a Scamex pelletizer. The composites created in this work will be referred to as PETGXOPP or PETGXMOPP, where “X” indicates the amount of additive in wt.%. The composites were then used as feedstock in an FGF Discovery 3D Granza machine purchased from Bárcenas CNC (Ciudad Real, Spain). Two types of plates were printed in the XY and XZ directions, according to ISO/ASTM 52921 (see Appendix A) [5,22]. Both plates were printed using a bead width of 2 mm, layer height of 1 mm, and printing speeds of 50 mm/s for the XY plates and 25 mm/s for the XZ plates. A constant temperature profile of 200/205/210 °C was established for all the materials studied after a temperature optimization process (see Appendix A). These temperatures correspond to the three heating zones of the extruder, the last one being the closest to the nozzle. In all cases, the platform temperature was set to 70 °C to ensure good adhesion of the first layer and avoid warping. Then, a LEKN(C1) 3020 CNC Router Machine Kit CNC was used to cut the specimens for tensile testing and thermal conductivity out of the printed plates, according to ISO 527 [23] and ISO 22007 [24] standards, respectively [5]. A 2 mm diameter flat milling cutter with two cutting edges was used to machine the specimens at a speed of 5000 rpm and a feed rate of 200 mm/min.

### 2.4. Characterization

The chemical structure of OPP, MOPP, and PBAT was analyzed by Fourier Transform Infrared Spectroscopy (FTIR) using a Bruker Alpha spectrometer (Billerica, MA, USA). The measurements were taken from 4000 to 650 cm^−1^ with a resolution of 4 cm^−1^. Contact angle measurements were conducted using deionized water on an FDM-printed goniometer. A coupled digital microscope was used to capture the images of the water droplets. The results of at least 3 independent measurements for each material were analyzed using the contact angle plugin of ImageJ software. The melt flow rate (MFR) was measured using a Lonroy LR-A001-A machine (Dongguan, China). The measurements were carried out by applying a load of 5 kg for 5 s at 210 °C for all the materials used in this work. The MFR of PETG was also measured at 245 °C as a control, following the recommended temperature given by the supplier. At least 3 independent measurements were performed to ensure the reproducibility of the results. The thermal conductivity of PETG and PETG15MOPP was measured in a DTC-25 conductivity meter (TA Instruments, New Castle, DE, USA) according to the ASTM E1350 standard. Modulated Differential Scanning Calorimetry (MDSC) analysis of PETG, OPP, and MOPP was carried out in a Q20 calorimeter device from TA Instruments (New Castle, DE, USA). Following a previously established protocol, all the samples were first heated to 160 °C for 10 min and then cooled down to 20 °C. Then, an oscillatory temperature sweep was performed from 20 to 300 °C at a rate of 1 °C/min with an amplitude of ±1 °C and a period of 120 s under nitrogen flow. The reversing heat flow values were extracted from these sweeps and used to calculate the specific heat capacity (C_p_) of PETG, OPP, and MOPP at different temperatures. The thermal stability of the materials was examined by thermogravimetric analysis (TGA) in a Q50 (TA Instruments, New Castle, DE, USA). Following a typical procedure, a temperature sweep from room temperature to 600 °C was performed using a constant rate of 10 °C/min. All the TGA experiments were carried out under a constant nitrogen flow. The mechanical characterization of the printed specimens was assessed by tensile testing in the Shimadzu AGS-X machine (Kyoto, Japan) using a constant speed of 2 mm/s. At least 5 specimens of each material were tested in all cases, in agreement with ISO 527. The morphology of OPP and MOPP and the fracture surface of the XY tensile tested specimens were examined by scanning electron microscopy (SEM) on a FEI Nova NanoSEM 450 microscope (Hillsboro, OR, USA) equipped with a field emission gun. Specimens were previously coated with a 10 nm Au layer in a Balzers SCD 004 Sputter Coater (Schaumburg, IL, USA).

## 3. Results and Discussion

OPP is described (like most wood-based materials) as a combination of lignin, cellulose, and hemicellulose. Lignin is a polysaccharidic polyphenol complex, showing an abundance of exposed aliphatic and aromatic hydroxylic moieties. Parallelly, cellulose and hemicellulose display a large abundance of primary and secondary alcoholic functionalities. In order to proceed with surface modification, it was assumed that, overall, OPP is a surface-hydroxylated structure and treated as such. Therefore, the surface-modification strategy was planned as depicted in Figure 2. First, OPP underwent carboxylation by reaction with chloroacetic acid in an alkaline environment, which led to the carboxymethylation of its nucleophilic OH groups, as previously reported for both lignin [25,26] and cellulose [25,27]. Then, carboxymethylated OPP was reacted with methanol to produce the methyl ester of carboxymethylated OPP via a common Fischer’s esterification reaction. This step was crucial to prepare the OPP surface for the grafting of PBAT, which was performed by tin-catalyzed polytransesterification of the dimethyl esters of terephthalic and adipic acids with 1,4-butanediol, adapting a procedure recently described for linear polyesters [28]. With this approach, DBTO can activate methyl ester moieties (both on the free monomer and the OPP surface) towards the nucleophilic attack of primary OH groups of 1,4-butanediol, leading to the formation of surface-grafted PBAT by the elimination of methanol by distillation.

The obtained MOPP was then characterized to assess the outcome of the surface modification. At first, FTIR confirmed the effectiveness of the PBAT grafting (Figure 3). In fact, MOPP displays all the main IR absorption bands typical of both OPP and PBAT. In particular, the broad O-H stretching band between 3000 and 3800 cm^−1^, typical of polyhydroxylated structures such as lignin and cellulose, has decreased in intensity, suggesting the presence of a smaller number of OH functionalities, in agreement with the effective grafting of PBAT on such functionalities. The intense C=O stretching peak of PBAT ester groups at 1720 cm^−1^ hides the corresponding vibration mode of lignin’s carbonyls as well as the stretching mode for highly symmetric 1,4-disubstituted aromatic rings. However, the shoulder peak at 1580 cm^−1^, present in the FTIR features of OPP, is still clearly visible in the spectrum of MOPP and related to the C=C stretching of the differently substituted aromatic rings of the polyphenols that compose the lignin structure [29,30]. Leaving out the overcrowded region between 800 and 1500 cm^−1^, in which specific features typical of OPP or PBAT cannot be identified separately, in the spectrum of MOPP, it is possible to identify a sharp peak at 745 cm^−1^ assigned to the vibration modes of para-disubstituted aromatic rings in the PBAT structure [31]. On the other hand, a broad peak between 450 and 750 cm^−1^, visible for both OPP and MOPP, is a typical fingerprint of the cellulose structure, as previously reported [32,33]. These results were supported by contact angle measurements of OPP and MOPP. Water droplets deposited on OPP rapidly spread on the surface, reaching contact angle values below 15 deg within seconds. In contrast, MOPP exhibited superhydrophobic behavior, with average values of 150 ± 9 deg.

The morphology of OPP and MOPP was analyzed by SEM. Both samples exhibit mostly elongated shapes, although some particles present a more spherical morphology (Figure 4a,b). The particle size distribution of both materials is similar, with an average lateral size of 75.0 ± 52.8 µm for OPP and 73.9 ± 44.5 µm for MOPP. Although the average particle size is 70–80 µm in both cases, some larger particles with sizes of hundreds of microns are also present (Appendix A). The surface of OPP looks fibrous, typical of agro-waste and biomass in general [12] (Figure 4c), whereas MOPP has a rougher appearance (indicated by yellow arrows in Figure 4d). This roughness, not visible in OPP, is associated with the PBAT coating [17,34].

The PETG composites were compounded in a twin-screw extruder with OPP or MOPP, using additive concentrations ranging from 5 to 15 wt.%. The pellets produced showed a homogeneous brown color, characteristic of the olive pit and no issues were encountered during the preparation of any of these materials. FGF printing was then carried out with these composites and PETG. A first test printing PETG5OPP at 240 °C, the recommended printing temperature by the manufacturer for PETG. However, the composite showed degradation signs at this temperature. Therefore, the printing temperature was decreased in intervals of 10 °C until the material flowed properly through the printing nozzle and the panels could be adequately printed at 210 °C. An additional test at a printing temperature of 205 °C increased the viscosity of the material too much, causing jams when printing PETG (see Appendix A for more details). The rest of the composites were also successfully printed at 210 °C, as well as PETG. Although the printing temperature is the same in all cases, the OPP composite plates presented a more irregular finish than those printed with the MOPP composites. This is clearly observed in the digital pictures of PETG15OPP and PETG15MOPP shown in Figure 5, where the high amount of OPP favors the formation of some agglomerates during printing, which are not big enough to clog the nozzle or affect drastically to the printing performance but increases the surface roughness of the printed plates.

This effect is not observed in the PETG15MOPP composite, likely because the surface modification allows for a better distribution of the MOPP within the PETG matrix. This did not cause any issues during the machining of the specimens for tensile testing or other experiments. Digital pictures of the XY- and XZ-printed plates of all the materials used in this work are shown in Appendix A.

The MFR of the different composites was then studied to better understand the differences observed in the surface roughness of the composites, aiming to correlate this with their processability (Figure 6). The values obtained for the OPP composites are slightly above 10 g/10 min. This value is in the lower limit recommended in the literature for processing materials by FGF to avoid clogging problems in the printer nozzle caused by the high viscosity of the melt. Similarly, the MFR of PETG at 210 °C is also rather low, close to 10 g/10 min as well. This value increases above 30 g/10 min when tested at 245 °C (the recommended temperature for printing PETG), showing a higher flowability at this temperature, more adequate for FGF [6,35]. The MFR values of the MOPP composites are like those of PETG tested at 245 °C. This is likely caused by the coating of the MOPP, which consists of a low molecular weight PBAT with a chemical composition very similar to that of the PETG matrix. These two factors probably favor the diffusion of the MOPP within the PETG chains, decreasing the overall viscosity of the melt. Improved flowability of the material (i.e., higher MFR) facilitates better deposition and interdiffusion of the polymer layers and strands, resulting in printed objects with a more uniform composition and better mechanical properties.

To explain the differences found in the MFR of the materials studied, the thermal properties of the composites (i.e., thermal conductivity and C_p_) were measured. The thermal conductivity of PETG15MOPP and PETG was first investigated from 3D-printed specimens expecting a higher thermal conductivity of the composite due to the presence of MOPP. However, the results obtained were rather similar for PETG and PETG15MOPP, obtaining values of 0.175 ± 0.024 W/m·K and 0.180 ± 0.014 W/m·K respectively. Then, MDSC tests were conducted on PETG, OPP, and MOPP to determine their C_p_ at different temperatures ranging from 50 °C to 300 °C. The reversing heat flow values for PETG, OPP, and MOPP are presented in Figure 7 and Appendix A. Figure 7a shows endothermic peaks in both OPP and MOPP, which are associated with the degradation of the agro-waste [36,37]. In the case of OPP, this peak occurs at a temperature of 188 ± 8 °C, with an associated enthalpy of 32 ± 17 J/g. For MOPP, this peak occurs at higher temperatures, 230 ± 3 °C, and with lower enthalpy (3 ± 1 J/g). These results suggest that the PBAT coating enhances the thermal stability of the MOPP, delaying and reducing its degradation. This also favors a more homogeneous distribution of the MOPP within the PETG matrix during the processing, having a higher viscosity of the melt (i.e., lower MFR), more suitable for FGF processing. The reversible heat flow data from the MDSC measurements were used to obtain the C_p_ values of PETG, OPP, and MOPP. Figure 7b) shows the C_p_ values averaged from 3 independent measurements at different temperatures between 150 and 220 °C in intervals of 10 °C. This graph shows that the C_p_ of OPP is higher than those of MOPP and PETG at temperatures below 180–200 °C. However, after degradation (i.e., when the endothermic peak is observed in Figure 7a and Appendix A), the C_p_ of OPP rapidly decreases, reaching values below those of MOPP.

TGA was performed to complete the thermal study of the materials (Figure 8a,b). The graphs show that both OPP and MOPP composites degrade earlier than PETG, indicating a lower thermal stability of OPP and MOPP than that of the polymer matrix [38,39]. However, MOPP exhibits superior stability than OPP, which extends to the composites and increases the onset of the maximum degradation temperature (Figure 8c,d). This enhanced thermal stability is in agreement with our results previously observed by MDSC and has been reported in the literature for agro-waste, whose compatibility with the matrix was improved through the use of additives [40].

The mechanical properties of the composites were studied by tensile testing. All the specimens tested were machined from the XY and XZ 3D-printed plates with good dimensional accuracy without encountering any issues during the cutting. Figure 9a,b present representative stress–strain curves for all the materials tested in this work. For clearer interpretation, Young’s modulus, tensile strength, and elongation at break values were extracted from all the specimens tested and are presented in Figure 10 as a function of the OPP or MOPP content (see Appendix A for more details).

In general, the elastic behavior of MOPP composites is superior to that of OPP composites and PETG, showing a higher Young’s modulus for both XY and XZ specimens (Figure 10a). This is because the MOPP is better integrated into the PETG matrix, reinforcing the material. On the other hand, the plastic behavior of both OPP and MOPP composites is lower than that of PETG in the XY orientation. This is due to the presence of agro-waste, which acts as a crack initiator and hinders the deformation of the polymer chains.

The tensile strength (Figure 10b) values decrease linearly for both OPP and MOPP composites of the XY-printed specimens, as the additives have worse mechanical properties than PETG. However, MOPP composites show higher strength than OPP composites. The results in the XZ orientation follow a different trend. Although the maximum strength also decreases in all composites, PETG5OPP shows higher strength than PETG5MOPP, likely due to the low concentration of additives, minimizing their aggregation in the PETG matrix. However, as the OPP content increases, the strength in the XZ orientation decreases linearly, while the strength values are maintained in the MOPP composites. The presence of both OPP and MOPP limits the movement of the PETG chains, resulting in material failure at much lower deformations in the XY specimens, as observed in the elongation at break values of Figure 10c). In this case, the surface modification with PBAT does not seem to enhance the ductility of the material, even when compared to OPP composites. This effect is also observed for the XZ specimens, where all materials tested break at values below 3% strain. While the OPP composites can withstand lower loads, they are able to deform to a larger extent, showing a certain plastic behavior. This is not observed in the case of the MOPP composites, which exhibit a brittle fracture when they break. This indicates that there must be a series of supramolecular interactions between PETG and the PBAT coating, which contribute to increasing the interlayer adhesion and improving the stiffness (Young’s modulus) and tensile strength of the MOPP composites, but it is not able to contribute to reaching larger deformations than the OPP composites or PETG [41].

These results align with the trend observed in multiple studies of non-coated OPP composites, which report an increase in Young’s modulus as the amount of OPP is increased in PS, PLA, PP, PCL, and PVC matrices [16,38,42,43,44]. These studies also report a decrease in tensile strength and elongation at the break due to the poor mechanical properties of OPP and its low compatibility with polymer matrices. To address this issue, various authors have modified the OPP surface through benzoylation or by using maleic anhydride to increase its hydrophobicity. In these cases, the mechanical properties improve when compared to the unmodified residue but do not surpass those of the polymer matrix [16,17], like the results obtained in this study.

Sinha et al. [45] developed a model to determine the interlayer strength of different structures, where C_p_ was considered a factor that quantifies the energy stored in the bead at the time of deposition. Furthermore, Compton et al. [46] observed that higher C_p_ values increase the energy rate in LFAM systems. This translates into higher temperatures during deposition (i.e., after leaving the printer nozzle) and lower viscosity (as observed in the MFR experiments), favoring material diffusion between beads and generating stronger bonds. This results in better interlayer bonding, as observed in MOPP composites, which exhibit higher C_p_ than OPP composites at 210 °C and also have higher tensile strength values in general.

SEM images of the fracture surface of the composites were captured to better understand their mechanical behavior. Low-magnification SEM images illustrating the fracture surfaces of the XY specimens are presented in Figure 11 and Appendix A. Voids between the layers and beads, typical of materials printed by FGF or FFF, visible in all samples, are indicated with blue arrows. These voids have average diameter sizes of 500 µm and are smaller in general in the MOPP composites than in PETG and the OPP composites. This is in agreement with the MFR results and the mechanical properties previously discussed since the MOPP composites have a lower viscosity of the melt, which favors a better interlayer bonding, reducing the gaps between layers and strands and also contributing to enhancing the mechanical properties.

On the other hand, the fracture surface of the OPP composites is full of smaller pores with diameters up to 300 µm. This porosity may have originated from two contributions: first, because of the poor compatibility of OPP with the PETG matrix; second, due to the accumulation of air and gases generated during printing due to OPP degradation that happens at temperatures above ca. 190 °C, as previously observed by MDSC. In contrast, the fracture surface of the MOPP composites, where MOPP degrades at a higher printing temperature, does not exhibit these porosities since MOPP exhibited a higher degradation temperature of ca. 230 °C, above the printing temperature used in this work (210 °C) [47,48]. Similar porosities have been documented in the literature for composites containing OPP in PP, PCL, and PLA matrices, reflecting poor compatibility between the matrix and additive [39,43,44,49,50]. These porosities also justify the lower mechanical properties observed in OPP composites compared to MOPP composites, including those of the XZ specimens, where the OPP composites had lower strength but higher elongation at break values, exhibiting a behavior similar to that of polymer foams [51,52].

This interaction of the OPP and MOPP with the polymer matrix is better illustrated in the high-magnification SEM images in Figure 12. The low-magnification images in Figure 11 showed very porous materials in the case of the OPP composites, but when the magnification is increased, the individual OPP particles can also be observed within this foamy structure. The OPPs also have gaps around them (indicated with yellow arrows in Figure 12a–c), demonstrating their poor integration into the PETG matrix [12,33,43]. In contrast, this does not occur in the MOPP composites, where the MOPP is well integrated within the PETG matrix, presenting a rougher morphology in the fracture surface. This proves why, in general, the MOPP composites presented higher mechanical properties than the OPP composites and also explains the higher stiffness obtained in some of the MOPP composites, which was even higher than that of the PETG matrix.

## 4. Conclusions

We have developed a series of composites for FGF-LFAM through the valorization of OPP/MOPP agro-waste, using PETG as a matrix. The OPP was modified by surface polymerization with PBAT to enhance its integration into the polymer, resulting in a hydrophobic material, MOPP. This modification provided polymer chains similar in nature to the PETG matrix, improving their compatibility. Consequently, these composites could be printed more easily using FGF due to a lower viscosity of the MOPP composites, as observed by MFR. MDSC and TGA studies indicated that the PBAT coating increased the degradation temperature of the MOPP and slowed its decomposition rate, preventing degradation during the printing process. This modification positively impacted the mechanical properties of the composites. Generally, the MOPP composites exhibited better mechanical properties than OPP composites in both XY and XZ printed specimens. While the mechanical properties of MOPP composites were still lower than those of the PETG matrix, the composites demonstrated the feasibility of valorizing up to 15 wt.% agro-waste. This reduces the amount of fossil fuel-derived plastic (PETG) and results in a material with higher stiffness than the polymer matrix, having values above 1600 MPa in the case of MOPP composites.

## Figures and Tables

**Figure 1 polymers-16-02981-f001:**
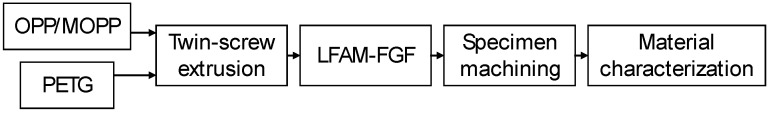
Manufacturing process of the composites developed in this work.

**Figure 2 polymers-16-02981-f002:**
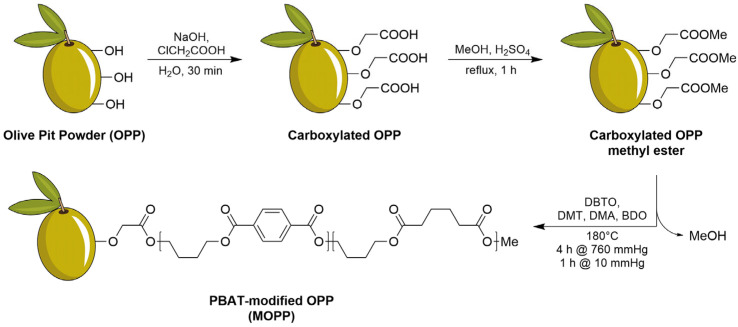
Surface modification strategy used for the grafting of PBAT from OPP surface. DBTO = dibutyltin (IV) oxide, DMT = dimethyl terephthalate, DMA = dimethyl adipate and BDO = 1,4-butanediol.

**Figure 3 polymers-16-02981-f003:**
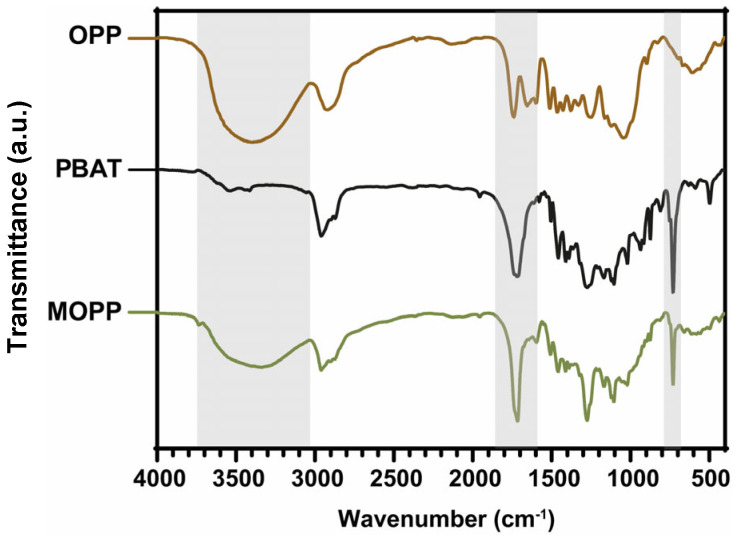
FTIR spectrum of OPP, MOPP and PBAT.

**Figure 4 polymers-16-02981-f004:**
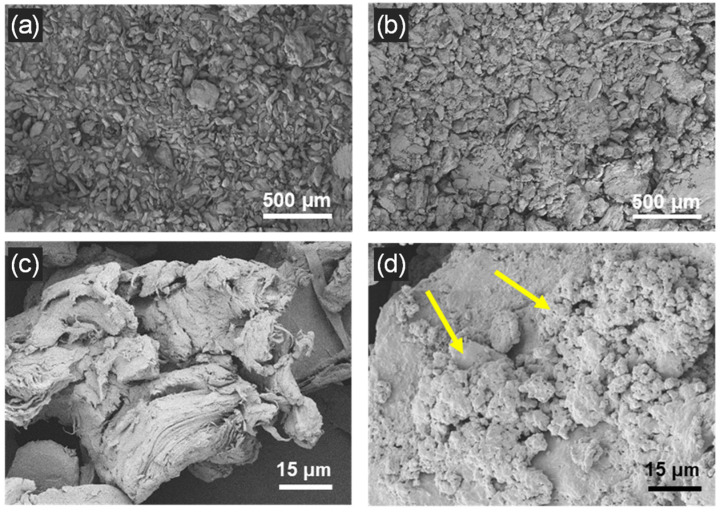
SEM images of (**a**) OPP; (**b**) MOPP; (**c**) detail of surface of OPP; and (**d**) detail of surface of MOPP.

**Figure 5 polymers-16-02981-f005:**
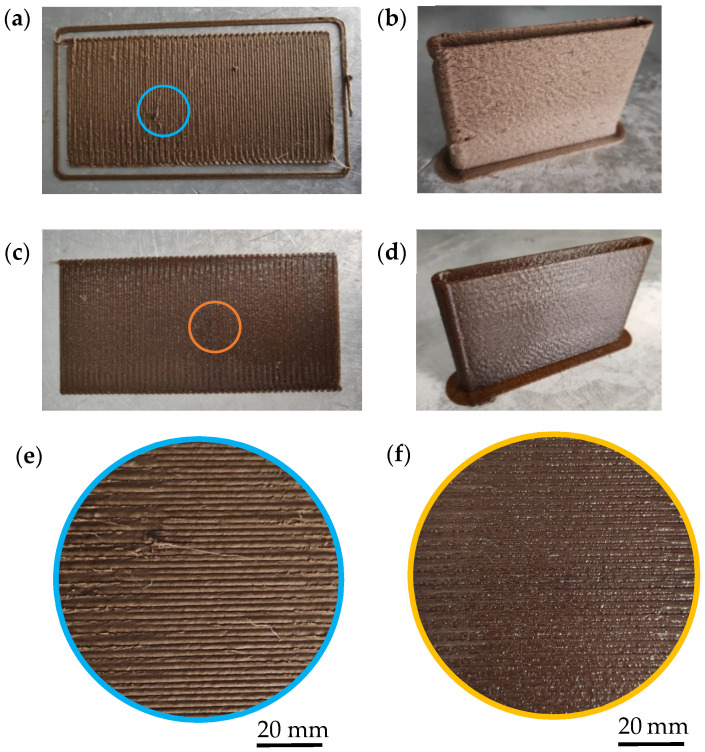
3D-printed (**a**) PETG15OPP XY plate; (**b**) PETG15OPP XZ plate; (**c**) PETG15MOPP XY plate; (**d**) PETG15MOPP XZ; (**e**) detail of PETG15OPP XY plate surface; and (**f**) detail of PETG15MOPP XY plate surface.

**Figure 6 polymers-16-02981-f006:**
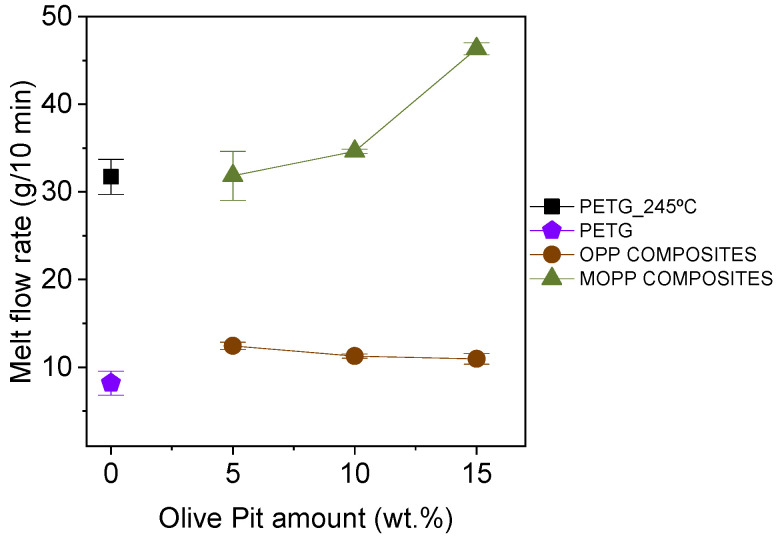
MFR of PETG, OPP composites and MOPP composites. The values were obtained at 210 °C unless otherwise specified.

**Figure 7 polymers-16-02981-f007:**
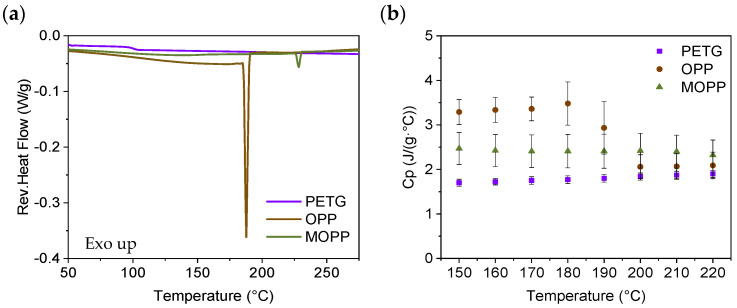
(**a**) MDSC curves and (**b**) C_p_ values for a range of temperatures between 150 and 220 °C for PETG, OPP, and MOPP.

**Figure 8 polymers-16-02981-f008:**
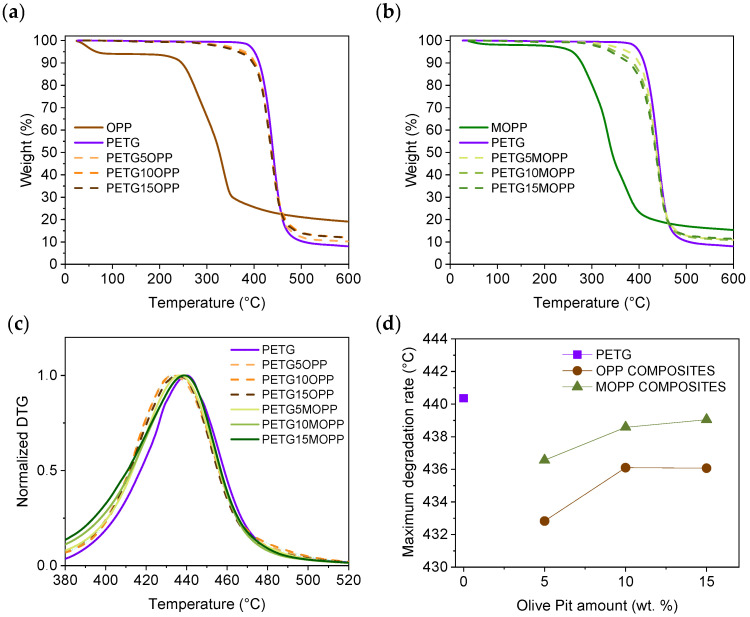
(**a**) TGA curves of OPP, PETG, and OPP composites; (**b**) TGA curves of MOPP, PETG, and MOPP composites; (**c**) DTG curves of PETG, OPP composites, and MOPP composites and (**d**) maximum degradation rate values of PETG, OPP composites, and MOPP composites.

**Figure 9 polymers-16-02981-f009:**
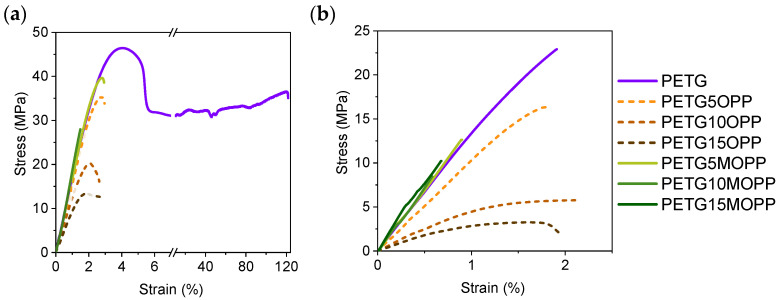
Representative stress–strain curves of (**a**) XY and (**b**) XZ printed specimens prepared by FGF.

**Figure 10 polymers-16-02981-f010:**
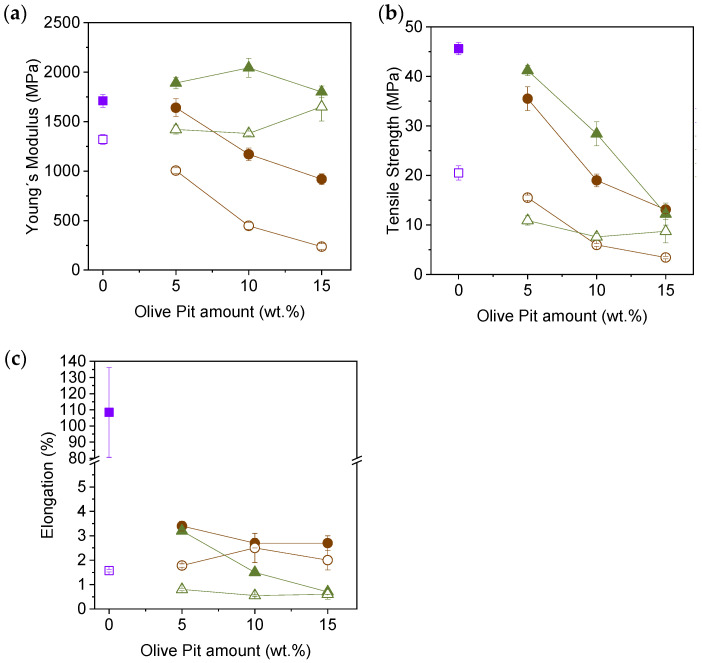
(**a**) Young’s modulus, (**b**) tensile strength, and (**c**) elongation at break values of printed PETG (purple), OPP composites (brown) and MOPP composites (green). Filled and howllow symbols represent XY- and XZ-printed specimens, respectively.

**Figure 11 polymers-16-02981-f011:**
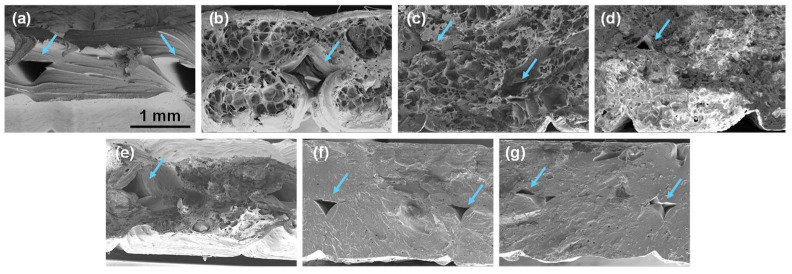
Low-magnification SEM images of the fracture surface of XY tensile testing specimens of (**a**) PETG; (**b**) PETG5OPP; (**c**) PETG10OPP; (**d**) PETG15OPP; (**e**) PETG5MOPP; (**f**) PETG10MOPP, and (**g**) PETG15MOPP. Blue arrows indicate voids between layers and beads. Scale bar in (**a**) is applicable to all images.

**Figure 12 polymers-16-02981-f012:**
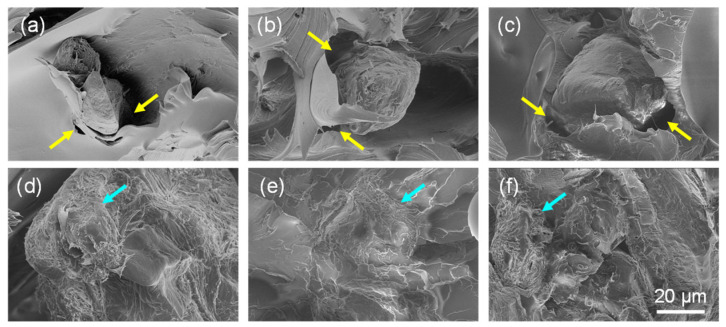
High magnification SEM images showing the detail of the fracture surface of the tensile testing specimens of (**a**) PETG5OPP; (**b**) PETG10OPP; (**c**) PETG15OPP; (**d**) PETG5MOPP; (**e**) PETG10MOPP, and (**f**) PETG15MOPP. Yellow arrows indicate the gaps between OPP and PETG while blue arrows indicate the presence of MOPP well integrated in the polymer matrix. Scale bar in (**f**) is applicable to all images.

## Data Availability

Data are contained within the article and Appendix A.

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
