# Peer review of "Development of Polymer Composites Using Surface-Modified Olive Pit Powder for Fused Granular Fabrication"

_polymers, 2024, doi:10.3390/polym16212981_

Round 1
Reviewer 1 Report
Comments and Suggestions for Authors
The work "Development of Polymer Composites Using Surface-Modified Olive Pit Powder for Fused Granular Fabrication" speak about the possibility to use olive pit powder for filler in a PETG matrix.
However given the poor compatibility between matrix and filler the latter was functionalized with PBAT.
The results are intresting and well displayed but something must be revised by the authors.
At line 90 appears number 12 at the end of the sentence. Is it a reference?
At page 5 the graphic of FTIR doesn't show a unit of measurement in ordinate. I suggest the authors include it.
At page 8 in my opinion the explanation provided by the authors on the increase of flowability of MOPP composite is not clear. If a filler has greater compatibility with the matrix theoretically the mechanical properties as well as the viscosity should increase. As the authors explain, however, the decrease in viscosity corresponds to an increase in workability?
Author Response
Reviewer 1
The work "Development of Polymer Composites Using Surface-Modified Olive Pit Powder for Fused Granular Fabrication" speak about the possibility to use olive pit powder for filler in a PETG matrix.
However given the poor compatibility between matrix and filler the latter was functionalized with PBAT.
The results are interesting and well displayed but something must be revised by the authors.
At line 90 appears number 12 at the end of the sentence. Is it a reference?
The reviewer is right. This is a reference of a previous work with more details about the sieving and grinding of the OPP.
At page 5 the graphic of FTIR doesn't show a unit of measurement in ordinate. I suggest the authors include it.
We have added the Y axis to Figure 2, as requested.
At page 8 in my opinion the explanation provided by the authors on the increase of flowability of MOPP composite is not clear. If a filler has greater compatibility with the matrix theoretically the mechanical properties as well as the viscosity should increase. As the authors explain, however, the decrease in viscosity corresponds to an increase in workability?
A higher compatibility of the MOPP with the matrix ensures better distribution (more homogeneous) during the processing of the composite when molten, preventing agglomeration issues in the printer nozzle that could otherwise increase the material's viscosity (i.e., lower MFR). When we refer to lower viscosity improving processability, we mean that the melt flows more easily, promoting better interlayer diffusion, which in turn leads to improved mechanical properties and a more homogeneous composition in the printed object. We have clarified this in the main text accordingly.
Reviewer 2 Report
Comments and Suggestions for Authors
The paper can be accepted after the minor revisions stated below.
- Please provide combined references such as [1-4] one by one and in detail, explaining their relationship to each other.
- Please add a flowchart of this study.
- Add figurese for chapters 2.2 - 2.3 - 2.4 and 2.5 to understand easily. (Used Equipments. for each step. etc).
- More details about SCAMEX twin-screw extruder. How did you gain the material give more detail.
- I would like to see test specimens. Before the tests and after the test.
Author Response
Reviewer 2
The paper can be accepted after the minor revisions stated below.
- Please provide combined references such as [1-4] one by one and in detail, explaining their relationship to each other.
We have revised the first paragraph of the introduction and have separated all the references for a clearer interpretation.
- Please add a flowchart of this study.
We have included a scheme depicting the whole manufacturing process of the composites in Figure 1.
- Add figurese for chapters 2.2 - 2.3 - 2.4 and 2.5 to understand easily. (Used Equipments. for each step. etc).
We believe that including images of all the equipment used would not significantly contribute to the understanding of the paper and would make it unnecessarily long. All the equipment used is commercially available, and we feel that it is adequately described to ensure that any researcher can reproduce the results presented in this work. We also did not include images of the equipment in our previous studies of a similar nature (e.g. A. Sanz de León et al. ACS Appl. Mater. Interfaces (2024), 16, 35554−35565; P. Burgos Pintos et al. Polymers (2024), 16, 60; and P. Burgos Pintos et al. Virtual and Physical Prototyping (2024), 19, 2386106).
- More details about SCAMEX twin-screw extruder. How did you gain the material give more detail.
We have rewritten this information in the Materials and methods section including the extruder used, the extruder speed, the L:D ratio, and the temperature profile used. We believe that this information is enough to ensure the reproducibility of our results, but we kindly ask the reviewer to tell us if there is any other information that must be included.
- I would like to see test specimens. Before the tests and after the test.
Unfortunately, we don’t have any pictures of the specimens. However, we can guarantee that all the specimens were machined adequately from the printed plates and in all cases at least 5 independent samples were tested according to the ISO 527 standard. We have included a sentence about this on page 9.